# Structured Assessments: Enhancing Success in Early Nursing Education and Student Retention

**DOI:** 10.3390/nursrep15090335

**Published:** 2025-09-11

**Authors:** Esther O. Park, Kathleen Chang, Susan Koduah

**Affiliations:** School of Nursing (SoN), College of Public Health (CPH), George Mason University (GMU), Fairfax, VA 22030, USA; kchang12@gmu.edu (K.C.); skoduah@gmu.edu (S.K.)

**Keywords:** nursing education, student retention, structured assessment, faculty consistency, sophomore attrition, nursing

## Abstract

**Background**: High attrition rates among nursing students, particularly during the sophomore year, threaten the sustainability of the nursing workforce. **Objectives**: This study assessed structured assessment methods implemented at a School of Nursing located in northern Virginia to improve student success and reduce attrition. **Methods**: Interventions included a three-tier grading system (Mastery, Intermediate, and Novice), standardized rubrics, detailed instructor guidelines, remediation sessions, and medication competency practice materials. **Results**: Data from student and instructor feedback surveys and attrition rate comparisons between Spring 2024 and Spring 2025 revealed a reduction in overall attrition of sophomores from 23.5% to 17.3% and from 20% to 12% in the Fundamentals courses. Students reported increased engagement and confidence in foundational core skills, though complex nursing skills care posed challenges. Instructors valued the structure but questioned the suitability of certain skills for sophomores. **Conclusions**: These findings underscore the value of structured assessments in enhancing retention, with implications for revision of curriculum design in students’ early years of nursing education.

## 1. Introduction

Nursing student attrition remains a persistent challenge globally, with significant implications for healthcare systems already strained by workforce shortages [1,2]. In the United States, attrition rates in prelicensure nursing programs range from 15% to 30%, with sophomore students, those in their second year, disproportionately affected, often due to academic failure in foundational courses [3,4]. These programs vary in structure: some are traditional four-year Bachelor of Science in Nursing (BSN) programs, where the sophomore year typically marks the beginning of intensive nursing course work, while others follow a 2 + 2 model, consisting of two years of prerequisite general education followed by two years of nursing-specific education, with the ‘sophomore’ phase often aligning with the first year of the actual nursing component. This period is pivotal as students transition from theoretical coursework to hands-on clinical and lab-based learning, requiring mastery of foundational skills like patient assessment, infection control and isolation, and medication administration [5,6]. The sophomore year often serves as a “make-or-break” phase, where academic rigor, emotional stress, and skill acquisition demands converge, leading to elevated dropout rates [7,8].

Globally, similar patterns emerge. In Namibia, Amakali et al. [9] reported that sophomore nursing students face unique barriers, including inadequate preparation for clinical demands, contributing to delayed completion or withdrawal. In Sweden, Lindberg et al. [10] found that the shift to professional identity formation during the second year heightens attrition risks. At the Primary Investigator’s (PI’s) School of Nursing, Spring 2024 data reflected an overall attrition rate of 23.2%, with 20% of attritions linked to the Fundamentals didactic/lab courses. This course, a cornerstone of the sophomore curriculum, introduces critical thinking and clinical judgment skills essential for nursing practice [6,11]. Its high attrition rate signals a need for targeted interventions to support student persistence.

Structured assessment methods offer a promising solution. Rooted in evidence-based pedagogy, these methods emphasize clarity, consistency, and support to enhance learning outcomes [12,13]. Components such as tiered grading systems, standardized rubrics, and faculty guidelines have been shown to reduce student anxiety, improve skill mastery, and foster retention [14,15]. Moreover, fostering a growth mindset, where students view challenges as opportunities for development, can mitigate the fear of failure often cited in attrition studies [16,17]. This study evaluated a multifaceted intervention package designed to reduce sophomore attrition and enhance engagement in the nursing fundamental courses, addressing both local and global challenges in nursing education.

## 2. Methods

This study was conducted during the Spring 2025 semester at the PI’s School of Nursing within the Fundamental didactic and lab courses, targeting sophomore students (*n* = 133). As an educational quality improvement initiative, it was exempt from Institutional Review Board (IRB) review as a study with human subjects (approved IRB ID: STUDY00000626). A mixed-methods approach was employed, integrating quantitative attrition data and survey responses with qualitative feedback.

### 2.1. Interventions

The interventions were developed by the research team, led by the principal investigator (PI) and including the co-authors, over a three-month period prior to the Spring 2025 semester, with input from the adjunct clinical lab instructors to ensure feasibility and buy-in. This process involved a thorough literature review, multiple design meetings, and iterative revisions based on instructor feedback. Key efforts included designing the structured Canvas course shell and orientation videos, creating standardized rubrics and detailed instructor guidelines with time schedules for labs (requiring approximately 32 h), and developing medication competency practice materials. To promote instructor buy-in, the research team conducted a dedicated two-hour orientation session to train them on implementation, further solidifying their commitment and ensuring consistent application across sections. Additionally, faculty members who spent time in after-hours remediation sessions were compensated at a competitive hourly rate from the PI’s grant.

A comprehensive set of interventions was implemented to ensure consistency, transparency, and student support:Three-Tier Grading System: Departing from traditional pass/fail metrics, the three-tier grading rubric, including a Mastery, Intermediate, and Novice level, was introduced. Adapted from Guskey and Link [14], this system evaluates the following competency:
○Mastery: Demonstrates full proficiency, no remediation needed.○Intermediate: Shows partial proficiency, benefits from practice.○Novice: Lacks basic proficiency, requires structured remediation. For example, a student achieving Mastery in Foley catheter insertion demonstrates sterile technique, patient communication, and documentation without errors. This approach aligns with growth-oriented learning theories [16].Critical Thinking Prompts: Instructors integrated prompts to enhance clinical reasoning, such as “What complications might arise if this step is omitted?” or “How would you prioritize care in this scenario?” These align with Tanner’s [18] clinical judgment model, encouraging reflective practice.Standardized Rubrics: Detailed rubrics were developed for each skill, specifying performance criteria. The indwelling catheter insertion/removal rubric, for instance, awarded points for hand hygiene (20%), sterile field maintenance (30%), catheter placement/removal (30%), and documentation (20%). Rubrics were distributed weekly, promoting transparency [19].Structured Course Shell: The Canvas platform featured orientation videos, weekly modules with readings and videos, and a detailed calendar. Students completed a syllabus agreement quiz to unlock content, reinforcing accountability [20].Instructor Guidelines: The adjunct faculty who teach different lab sections received hourly guidelines to standardize lab delivery. For example, in Week 1, 30 min were allocated to PPE demonstration, 45 min to practice, and 15 min to assessment, ensuring uniformity across sections [21].Medication Competency-Dosage Calculation Practice: Weekly problem sets (10 questions) progressed from basic conversions (e.g., mg to g) to complex infusion rates, building math competency critical for patient safety [22].Remediation Sessions: Optional after-hours sessions targeted students rated Novice, students’ volunteering, or the instructors’ discretion, offering one-on-one coaching and practice. For instance, a student struggling with wound care could revisit sterile technique with an instructor during the after-hours [23].

### 2.2. Sampling and Procedure

Participants for the intervention group were purposively sampled from sophomore cohorts enrolled in Fundamentals courses during Spring 2025 (*n* = 133), encompassing all students in the cohort to ensure comprehensive implementation of the interventions. To evaluate the effectiveness, a historical control was utilized by comparing outcomes with the Spring 2024 sophomore cohort (*n* = 85), which did not receive any of the structured assessment interventions and thus served as the control year for baseline comparison. The 2024 cohort was selected retrospectively based on available archival data from the same Fundamental courses, with no additional selection criteria applied beyond enrollment in the courses during that semester. Interventions were piloted and implemented across all lab sections in Spring 2025, with faculty trained via a two-hour orientation session to ensure consistent delivery. Data collection for the 2025 intervention group spanned 4 weeks, aligning with the semester lab timeline, while 2024 data were retrospectively analyzed primarily for attrition rates to enable direct year-over-year comparisons.

Interventions were piloted in Spring 2025 across all lab sections, with faculty trained via a two-hour orientation. Data collection spanned 12 weeks, aligning with the semester timeline.

### 2.3. Data Collection and Analysis

Data sources included the following:

Student Surveys: Weekly 5-point Likert-scale surveys (*n* = 120, 90% response rate) assessed confidence and engagement, supplemented by open-ended questions.

Instructor Surveys: End-of-lab session surveys (*n* = 8) evaluated intervention effectiveness.

Attrition Rates: Spring 2024 and 2025 data were compared.

Quantitative data underwent descriptive statistical analysis using SPSS v.27, while qualitative responses were thematically coded following Braun and Clarke’s [24] framework.

## 3. Results

### 3.1. Student Feedback

Surveys (*n* = 120, 90% response rate) revealed strong engagement with foundational skills but challenges with complexity. In Week 1, 92% of students felt confident in Personal Protective Equipment (PPE) and hand hygiene, citing clear rubrics as helpful, while 35% identified bed bath as their least confident skill due to patient positioning difficulties. Week 2 showed 88% confidence in sterile gloving and 86% in Foley catheter insertion, with 45% reporting wound care as challenging due to its multistep nature. By Week 3, 83.8% felt confident in pre-op care and SBAR communication (See Table 1), but 50% struggled with post-op care, often citing time pressure during validation.

Qualitative feedback highlighted the rubrics’ value, with 78% agreeing they clarified expectations. However, 32% felt overwhelmed by Week 3’s pace, suggesting a need for additional lab weeks allocated for the multiple complex nursing skills.

### 3.2. Instructor Feedback

Instructors (*n* = 8) identified PPE (57.1%) and patient identification (28.6%) as Week 1 strengths, consistent with student reports. In Week 2, student performance excelled in sterile gloving (85.7%) and indwelling (Foley) catheter insertion (57.1%), while it lagged in perineal care (28.6%). Week 3 feedback was limited (87.5% missing responses), but pre-op checklists were praised.

Instructors lauded the Canvas structure and guidelines for reducing variability, with one noting, “The hourly guidelines kept us on track.” However, 50% questioned the appropriateness of advanced skills (e.g., Foley insertion) for sophomores, recommending their deferral to junior year or requiring more weeks for lab allocation.

### 3.3. Remediation Sessions

Feedback from remediation sessions (*n* = 4) indicated strong student satisfaction. All participants (100%) strongly agreed that the sessions provided opportunities to practice desired skills and felt supported. Additionally, 100% strongly agreed that the lab deepened their knowledge from didactic courses and that they learned nursing skills effectively. Qualitative feedback highlighted the value of one-on-one learning and the positive impact on skill confidence.

### 3.4. Attrition Rates

Attrition decreased significantly. Spring 2024 showed an overall rate of 23.5% (20/85 students), with 20% (17/85) from the fundamental course. In Spring 2025, overall attrition fell to 17.3% (23/133), and the fundamental course’s attrition dropped to 12% (16/133), yielding reductions of 6.2% and 8%, respectively (See Table 2). These reductions indicate the efficacy of the interventions in supporting retention.

## 4. Discussion

The structured interventions effectively reduced attrition and enhanced student engagement, aligning with evidence that consistency and transparency are key to nursing student success [2,4,17]. The three-tier grading system, in particular, shifted the focus from punitive measures to growth and remediation, reflecting Lewis et al.’s [16] concept of a growth mindset. This approach likely mitigated the fear of failure, a known contributor to attrition [8,25]. Standardized rubrics and detailed instructor guidelines further reduced subjectivity in assessments, addressing a common student grievance in clinical education [15].

However, the persistent challenges with complex skills like post-operation care and wound care suggest a potential mismatch between skill difficulty and readiness in the early years of nursing. This finding echoes Mani’s [26] assertion that nursing curricula must be developmentally appropriate to avoid overwhelming students. Instructors’ concerns about skill appropriateness reinforce this, suggesting that advanced skills might be better suited for later years when students have a stronger clinical foundation, or the current short time period of lab weeks should be extended to make nursing skills evenly distributed. Based on the study results and faculty feedback, planned curriculum changes include transferring specific advanced skills, such as Indwelling (Foley) catheter insertion, wound care, and post-operative care, from the sophomore Fundamental courses to the junior year. This adjustment would allow sophomores to consolidate foundational competencies, including PPE donning/doffing, hand hygiene, patient identification, and sterile gloving, before progressing to more intricate procedures. Additionally, extending laboratory sessions by one to two weeks for the remaining sophomore skills could provide more practice time, reducing overload and enhancing mastery. Future curriculum revisions could delay these skills or introduce them gradually, perhaps through simulation-based learning, which has been shown to build confidence before live practice [11].

The study’s findings also align with broader trends in nursing education. As programs increasingly adopt competency-based models, structured assessments like those implemented here could serve as a blueprint for standardizing skill evaluation across diverse settings [3,26]. Moreover, the integration of dosage calculation practice as part of medication competency addresses a critical gap, as medication errors remain a leading cause of adverse events in healthcare [22]. By embedding math competency early, this intervention not only supports student success but also enhances patient safety.

Recent trends in nursing education have involved reducing the number of weeks dedicated to nursing skills laboratory training, allowing students to spend more time in clinical sites like hospitals. However, securing adequate clinical placement has become increasingly difficult due to factors such as insufficient availability in healthcare facilities, shortages of nurse educators, and competition among programs. As a result, skills laboratories play a vital role in replacing some clinical time to ensure students receive the necessary hands-on experience [27]. Nevertheless, a study by Saghafi et al. [28] demonstrated that nursing skills lab-based learning activities significantly improve nursing students’ critical thinking skills. This finding suggests that extending the duration of skills laboratory training could be beneficial for enhancing students’ critical thinking and clinical judgment skills.

## 5. Limitations and Future Research

While promising, this study has limitations. Its single-site design limits generalizability, and reliance on self-reported data may inflate perceived success due to social desirability bias [29]. Additionally, the high rate of missing instructor feedback in Week 3 (87.5%) weakens conclusions about later skills. Future research should include multi-site evaluations with objective metrics, such as nursing skill pass rates or standardized test scores, to validate these findings. Longitudinal studies tracking students’ clinical performance post-graduation could also assess the long-term impact of structured assessments on workforce readiness.

## 6. Conclusions

This study demonstrates that structured assessment methods can significantly reduce sophomore nursing student attrition and enhance engagement. By providing clear expectations, consistent instruction, and targeted support, these interventions address key barriers to student success. However, the challenges with complex nursing skills introduced in their early year of nursing education highlight the need for ongoing curriculum refinement to ensure developmental appropriateness. As nursing education evolves to meet global workforce demands, structured assessments offer a scalable, evidence-based strategy to strengthen the nursing pipeline. Continued innovation and evaluation are essential to optimize outcomes for both students and the healthcare systems they will serve.

## Figures and Tables

**Table 1 nursrep-15-00335-t001:** Student confidence in weekly skills.

Week	Most Confident Skills	% Confident	Least Confident Skills	% Least Confident
1	PPE, hand hygiene	92%	Bed bath	35%
2	Sterile gloving, Foley catheter	88%, 86%	Wound care	45%
3	Pre-op care, SBAR communication	83.8%	Post-op care	50%

**Table 2 nursrep-15-00335-t002:** Attrition rate comparison.

Semester	Overall Attrition	The Course Attrition
Spring 2024	23.5% (20/85)	20% (17/85)
Spring 2025	17.3% (23/133)	12% (16/133)

## Data Availability

Data available upon request from the corresponding author.

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
