# Peer review of "Structured Assessments: Enhancing Success in Early Nursing Education and Student Retention"

_nursrep, 2025, doi:10.3390/nursrep15090335_

Round 1
Reviewer 1 Report
Comments and Suggestions for Authors
Structured Assessments: Enhancing Success in Early Nursing Education and Student Retention, submitted in Nursing Reports, is a well-structured, brief review that addresses an important topic in nursing: reducing dropout rates. The results are promising, and the interventions are clearly described. However, with minor corrections, especially regarding additional clarifications in methodology and discussion, it could be even better.
The title is clear and concise, and the keywords are well chosen.
The abstract is good, it includes the aim, interventions, results and conclusion. Eventually, the word "amendment" could be replaced with something clearer, such as "refinement" or "revision", to emphasise that additional improvement is needed, not just change.
The introduction is solid and provides good context for the problem, with the authors citing dropout rates and global trends, as well as specific issues at their own University, which lends local relevance to the study. Additionally, the transition from theory to practice in the second year is well emphasised, which is crucial in light of the high dropout rate.
The method is clear. It shows who the participants were, what interventions were implemented and how the data were collected. Seven interventions are described in detail. However, Sampling and Procedure could be a bit more detailed. For example, how was the control group selected (if any)? Although data from 2024 and 2025 are being compared, it could be more clearly stated that 2024 served as a control year.
The results are clearly presented in text and a single table, which is great.
Discussion is strong. It links the results to existing literature and theory. The authors confirm the student's difficulties with complex skills and propose solutions, such as postponing these skills to a later year or extending the duration of laboratory exercises. Perhaps more details could be added here on how exactly the curriculum is planned to be changed based on these results. For example, what skills exactly are considered for transfer to a later year of study?
Limitations and Future Research are realistic and honest. The authors recognise the limitations of the study and suggest relevant future research.
References are carefully selected and relevant.

Author Response
Structured Assessments: Enhancing Success in Early Nursing Education and Student Retention, submitted in Nursing Reports, is a well-structured, brief review that addresses an important topic in nursing: reducing dropout rates. The results are promising, and the interventions are clearly described. However, with minor corrections, especially regarding additional clarifications in methodology and discussion, it could be even better.
The title is clear and concise, and the keywords are well chosen. – We added one more keyword, nursing
The abstract is good, it includes the aim, interventions, results and conclusion. Eventually, the word "amendment" could be replaced with something clearer, such as "refinement" or "revision", to emphasise that additional improvement is needed, not just change. – Thank you for your feedback and we agree with this comment. Therefore, “amendment” is replaced by “revision.”
The introduction is solid and provides good context for the problem, with the authors citing dropout rates and global trends, as well as specific issues at their own University, which lends local relevance to the study. Additionally, the transition from theory to practice in the second year is well emphasised, which is crucial in light of the high dropout rate. -Thank you very much
The method is clear. It shows who the participants were, what interventions were implemented and how the data were collected. Seven interventions are described in detail. However, Sampling and Procedure could be a bit more detailed. For example, how was the control group selected (if any)? Although data from 2024 and 2025 are being compared, it could be more clearly stated that 2024 served as a control year. - Thank you for your invaluable feedback. This input has truly helped us improve the "Sampling and Procedure" section. The changes have been made on page 3 and are highlighted in red.
The results are clearly presented in text and a single table, which is great.
Discussion is strong. It links the results to existing literature and theory. The authors confirm the student's difficulties with complex skills and propose solutions, such as postponing these skills to a later year or extending the duration of laboratory exercises. Perhaps more details could be added here on how exactly the curriculum is planned to be changed based on these results. For example, what skills exactly are considered for transfer to a later year of study? - We appreciate your valuable feedback. It has significantly aided us in refining the "Discussion" section. The revisions appear on page 5 and are marked in red.
Limitations and Future Research are realistic and honest. The authors recognise the limitations of the study and suggest relevant future research. -Thank you very much. We appreciate your time.
References are carefully selected and relevant. -Thank you very much. We appreciate your time in reviewing this manuscript.
Reviewer 2 Report
Comments and Suggestions for Authors
Thank you for the opportunity to review this Brief Report which describes and effort to re-structure and standardize skills education in a nursing Fundamentals course in the sophomore year of a BSN nursing program. The authors present a strong case for the interventions, especially in terms of attrition of badly needed graduates in the sophomore year. The biggest concern is the ?outdated literature used, with the exception of the global data. The interventions are well described, albeit it is not known if they are original or gleaned from other literature. It is not known the level of effort required to design and implement these interventions. Student feedback was helpful, and showed an amazing response rate, including some qualitative data. The authors note that there was an end-of-semester instructor survey – yet the results are given for the first three weeks of the semester. Nevertheless, the results contribute to overall understanding and subsequent suggestions. In a brief report, there is a space restriction that precludes fulsome presentation, however, the discussion flowed well and was focused. The poor response rate of the instructors needs additional attention – it may be that the students are not the only ones who are overwhelmed. The absence of more recent literature at least needs to be mentioned/discussed. Some of the doi numbers are missing. A good effort to test in real time innovations in nursing education and share with other educators.
Brief Report – nursrep-3849978
As directed, here is an absolute list of strengths and weaknesses.
Strengths
- Topic – actually reported on interventions to alleviate attrition/failure in a specific course.
- Well written
- Tested the interventions with students and instructors
- Final results focused on attrition/failure rate attributable to the single course – fits the introduction.
Areas to Revise
- Citations used in introduction and throughout paper are quite dated – if there is no more recent literature, this needs to be stated e.g. results of a search.
- Described interventions well, but failed to account for time and effort to design and implement changes – e.g., CANVAS, design orientation, time schedules for labs etc. Did the researchers do this or the whole instructor group? Buy-in by instructors?
- Fix inconsistencies – e.g., reported an end of semester survey for instructors, but only reported on first three weeks, if survey not completed until end, how good are memories of the first few weeks? Also please explain poor instructor response – were they overwhelmed with additional work, etc.
- Reported amazing student response rate – how was this achieved? Ethics? Were the researchers the teachers of the course?
- Need to report that attrition in sophomore year is often due to failure in a course – implied by not supported by data.
- Doi numbers missing from some references – some because they are old, but some do have numbers that are not reported.
- Noticed keyword of “integrative review”. This does not meet the criteria for an integrative review.
Author Response
Reviewer 2
Thank you for the opportunity to review this Brief Report which describes and effort to re-structure and standardize skills education in a nursing Fundamentals course in the sophomore year of a BSN nursing program. The authors present a strong case for the interventions, especially in terms of attrition of badly needed graduates in the sophomore year. The biggest concern is the ?outdated literature used, with the exception of the global data. The interventions are well described, albeit it is not known if they are original or gleaned from other literature. It is not known the level of effort required to design and implement these interventions. Student feedback was helpful, and showed an amazing response rate, including some qualitative data. The authors note that there was an end-of-semester instructor survey – yet the results are given for the first three weeks of the semester. Nevertheless, the results contribute to overall understanding and subsequent suggestions. In a brief report, there is a space restriction that precludes fulsome presentation, however, the discussion flowed well and was focused. The poor response rate of the instructors needs additional attention – it may be that the students are not the only ones who are overwhelmed. The absence of more recent literature at least needs to be mentioned/discussed. Some of the doi numbers are missing. A good effort to test in real time innovations in nursing education and share with other educators.
Brief Report – nursrep-3849978
As directed, here is an absolute list of strengths and weaknesses.
Strengths
- Topic – actually reported on interventions to alleviate attrition/failure in a specific course.
- Well written
- Tested the interventions with students and instructors
- Final results focused on attrition/failure rate attributable to the single course – fits the introduction.
Areas to Revise
- Citations used in introduction and throughout paper are quite dated – if there is no more recent literature, this needs to be stated e.g. results of a search. - Thank you for your invaluable feedback. Based on your advice, we have eliminated several outdated citations and replaced them with more recent evidence. The new changes and additions to citations and references have been made throughout the paper and are marked in green. Any deletions are marked with strikethrough throughout the manuscript.
- Described interventions well, but failed to account for time and effort to design and implement changes – e.g., CANVAS, design orientation, time schedules for labs etc. Did the researchers do this or the whole instructor group? Buy-in by instructors? - Thank you for your valuable feedback; we agree with your assessment. Accordingly, we have added a more detailed explanation of the interventions, including the time and effort involved in their design and implementation, the roles of the research team and instructors, and steps taken to ensure instructor buy-in. These additions can be found on pages 2-3 of the revised manuscript and are highlighted in green.
- Fix inconsistencies – e.g., reported an end of semester survey for instructors, but only reported on first three weeks, if survey not completed until end, how good are memories of the first few weeks? – Thank you for this sharp catch: "end-of-semester" was changed to "end-of-lab" session. The lab session lasts for the first 4 weeks, including the 4th week's competency test and remediation. Thus, this doesn't impact biased memory. The change is found in page 4 in green text.
Also please explain poor instructor response – were they overwhelmed with additional work, etc. Most of the lab instructors were adjunct faculty, and they seemed to have less commitment to this type of survey when asked. Additionally, when we sent out the survey link, we followed research ethics and did not make the response mandatory. The following is the verbiage we used in that survey, and I believe the instructors chose not to respond at the end of the lab sessions: Your participation is entirely optional, and choosing not to participate will have no impact on your professional standing, evaluations, or relationship with the institution. If you begin the survey, you may withdraw at any time without consequence by simply closing the window. Please be assured that the survey is fully anonymous: no personally identifiable information (such as names, employee IDs, or IP addresses) is collected, and responses cannot be traced back to individuals. All data will be used solely for educational and programmatic improvement purposes and handled in accordance with ethical guidelines.
- Reported amazing student response rate – how was this achieved? Ethics? Were the researchers the teachers of the course? Thank you for your feedback. The study was quasi-experimental, more akin to a QI project aimed at improving teaching and learning. Research ethics were strictly followed, as evidenced by the message below, and we can attest that there was no coercion: You have now completed the Week XX N244 Lab. To help improve future labs and enhance the learning experience for students, I invite you to voluntarily share your feedback through this brief anonymous survey: The survey consists of 5 questions and should take only 1-2 minutes to complete. Your participation is entirely optional, and choosing not to participate will have no impact on your grades, standing in the course, or relationship with the instructor or institution. If you begin the survey, you may withdraw at any time without consequence by simply closing the window. Please be assured that the survey is fully anonymous: no personally identifiable information (such as student IDs, names, or IP addresses) is collected, and responses cannot be traced back to individuals. All data will be used solely for educational improvement purposes and handled in accordance with ethical guidelines.
We believe the high response rate was due to the short time the survey takes (less than 1 minute) and the reminders we sent out twice a day for three days, once each weekly lab was completed. This likely encouraged student to participate in the survey.
- Need to report that attrition in sophomore year is often due to failure in a course – implied by not supported by data. Thank you for your invaluable feedback. We agree and have made the corresponding changes. These changes can be found on page 2 in the "Introduction" section, where the revised text is highlighted in green.
- Doi numbers missing from some references – some because they are old, but some do have numbers that are not reported. Thank you for the thoughtful comment. We have added all available DOIs, and the newly added ones are highlighted in green under ‘References.’
- Noticed keyword of “integrative review”. This does not meet the criteria for an integrative review. – Thank you for the feedback. The keywords we submitted are: nursing education, student retention, structured assessment, faculty consistency, sophomore attrition, and nursing (included as requested this time). We are unsure why "integrative review" appears as a keyword, as it was not part of our manuscript. We would consult the editor to investigate any potential systematic error in the keyword display.
Reviewer 3 Report
Comments and Suggestions for Authors
This is a timely topic with increasing attention devoted to competency-based assessment and determining what works. IN your introduction, you will need to explain that some nursing schools are 4 years and some are 2+2.
Somewhere in the paper, you should include a section on how you staffed some of the interventions you suggest. What was the impact on faculty time?
On page 3, "Equipments" should be "Equipment."
On page 3, this sentence "In Week 2, sterile gloving (85.7%) and Foley catheter insertion (57.1%) excelled, while perineal care lagged (28.6%)" implies that skills lagged while it was student performance that did. This and the following sentence should be re-written.
On page 4 in the section on attrition rates, there are semi-colons where there should be commas.
On page 5, you mention a trend toward decreasing lab use. What needs to also be mentioned here is how difficult clinical placements are and the need for labs to replace some clinical time.
Some references are dated. For example, relying on a reference from 2007 and relating it to current issues is not appropriate. Lasater 2007 is cited on page 3 in the results section. Not sure why.
Author Response
Reviewer 3
This is a timely topic with increasing attention devoted to competency-based assessment and determining what works. IN your introduction, you will need to explain that some nursing schools are 4 years and some are 2+2. This feedback was very helpful in catching up on the notes about the program differences. The additional sentences we added can be found on page 2, and that text is in blue.
Somewhere in the paper, you should include a section on how you staffed some of the interventions you suggest. What was the impact on faculty time? This feedback was somewhat similar to the second reviewer's feedback point #2. In addition to our response to Reviewer 2, we added some more details on page 3 in the "Intervention" section, and the text is in blue.
On page 3, "Equipments" should be "Equipment." à The original page 3 has shifted to page 4; the typo was corrected accordingly on page 4.
On page 3, this sentence "In Week 2, sterile gloving (85.7%) and Foley catheter insertion (57.1%) excelled, while perineal care lagged (28.6%)" implies that skills lagged while it was student performance that did. This and the following sentence should be re-written. à Thank you for the feedback. The sentence has been rewritten and can now be found on page 5, and the revised text is in blue.
On page 4 in the section on attrition rates, there are semi-colons where there should be commas. à To avoid any confusion, that sentence was revised as follows: These reductions indicate the efficacy of the interventions suggest the interventions’ efficacy in supporting retention.
On page 5, you mention a trend toward decreasing lab use. What needs to also be mentioned here is how difficult clinical placements are and the need for labs to replace some clinical time. à Thank you for this helpful feedback. We have amended the section appropriately. The changes can now be found on page 6, and the revised text is in blue. To amend the section according to your feedback, one citation was added in-text and in the reference (text is in blue).
Some references are dated. For example, relying on a reference from 2007 and relating it to current issues is not appropriate. Lasater 2007 is cited on page 3 in the results section. Not sure why. à All references older than 10 years have been replaced with the most recent evidence. However, certain older references, such as Lasater (2007), have been retained as they are considered gold standards. As advised, the Lasater (2007) reference in the results section has been removed. The updated references are found throughout the paper and in the ‘reference’ section.